# The association between intersystem prison transfers and COVID-19 incidence in a state prison system

Lauren Brinkley-Rubinstein[1,2], Katherine LeMasters[2,3]*, Phuc Nguyen[4], Kathryn Nowotny[5], David Cloud[6], Alexander Volfovsky[4]

1 Department of Social Medicine, University of North Carolina at Chapel Hill, Durham, NC, United States of America, 2 Center for Health Equity Research, University of North Carolina at Chapel Hill, Durham, NC, United States of America, 3 Department of Epidemiology, University of North Carolina at Chapel Hill, Durham, NC, United States of America, 4 Department of Statistical Sciences, Duke University, Durham, North Carolina, United States of America, 5 Department of Sociology, University of Miami, Miami, Florida, United States of America, 6 Department of Behavioral Sciences and Health Education, Emory University, Atlanta, Georgia, United States of America

* katherine.lemasters@unc.edu

## Abstract

Prisons are the epicenter of the COVID-19 pandemic. Media reports have focused on whether transfers of incarcerated people between prisons have been the source of outbreaks. Our objective was to examine the relationship between intersystem prison transfers and COVID-19 incidence in a state prison system. We assessed the change in the means of the time-series of prison transfers and their cross-correlation with the time-series of COVID-19 tests and cases. Regression with automatic detection of multiple change-points was used to identify important changes to transfers. There were over 20,000 transfers between the state's prisons from January through October 2020. Most who were transferred (82%), experienced a single transfer. Transfers between prisons are positively related to future COVID-19 case rates but transfers are not reactive to current case rates. To mitigate the spread of COVID-19 in carceral settings, it is crucial for transfers of individuals between facilities to be limited.

## Introduction

Prisons are the epicenter of the COVID-19 pandemic in the United States, accounting for most of the largest single site cluster outbreaks in the pandemic [1]. Many factors and policies amplify the risk of COVID-19 infection and transmission, including movements or transfers of people to different facilities within a prison system [2]. Transfers of incarcerated people have been given much attention in the last year. In May 2020, the California Department of Corrections and Rehabilitation transferred 122 people from the California Institution for Men to its San Quentin facility. Within days, San Quentin experienced an outbreak that led to nearly one-third of all incarcerated people having COVID-19 and 28 deaths [3]. On February 1, 2021, the Office of the Inspector General released a report concluding, "California

**Data Availability Statement:** All data files are available from The COVID Prison Project.

**Funding:** LBR: The Robert Wood Johnson Foundation (78304) https://www.rwjf.org/ The

funders had no role in study design, data collection and analysis, decision to publish, or preparation of the manuscript.

**Competing interests:** The authors have declared that no competing interests exist.

Correctional Health Care Services and the California Department of Corrections and Rehabilitation caused a public health disaster at San Quentin State Prison when they transferred medically vulnerable incarcerated persons from the California Institution for Men without taking proper safeguards" [3].

Prison and jail officials have taken some measures to limit movements within and between institutions [4, 5]. The most pervasive examples include prohibiting visitation, placing housing units or entire facilities on lockdown, or placing individuals in quarantine or medical isolation. Intermittently, prison systems have stopped transfers proactively or reactively to mitigate an ongoing outbreak. While media accounts have detailed singular outbreaks that have occurred subsequent to transfers, no study has empirically examined the relationship between transfers and COVID-19 incidence. In this paper, transfer patterns of a large, southeastern state prison system in the United States are described in addition to the association between transfer patterns and COVID-19 cases in their prisons.

## Methods

This study included publicly available time-series data from one state with a mid-sized prison population. Data was comprised of COVID-19 cases and tests and transfers of incarcerated people from all prison facilities within one state prison system from January to October 2020. The unit of analysis is the state-wide time-series of prison transfers and COVID-19 cases. The change in the means of the time-series and their cross-correlations was analyzed. As all data were publicly available, this study was exempted from human subjects review by the Duke University Institutional Review Board.

### Data

COVID-19 incidence data were from the COVID Prison Project (CPP) [6]. The CPP publishes a state-level dataset with the number of daily COVID-19 tests, confirmed cases, and deaths by prison facility. The data are aggregated by CPP based on public reports by prison systems. The state prison system did not report COVID-19 tests or cases prior to April 2020. The state updated *cumulative* test and case numbers on a daily basis. To construct the dataset used in this paper the daily change in those reported cumulative numbers was computed. In the event that the cumulative numbers decreased, we assigned a value of zero to that day. Decreases in the cumulative values were not explained by the state. Transfer data are from the state's offender public information database and were linked by CPP. In this database, each movement (e.g., release, death, return from parole, intersystem transfers, new admissions) of every incarcerated person was recorded and reported weekly. Prison transfers include transfers between work release facilities but exclude admissions and releases to jails. Daily transfer was defined as the number of movements received from one prison and transferred to another prison within the state prison system.

### Statistical analysis

To analyze if and when transfer volume changed, a regression with automatic detection of multiple change-points to the means of the time-series was applied. Given a number of change points, the regression infers the locations of these change points to best fit a set of functions for segments of data separated by change points. A Bayesian implementation in package mcp was used to find the step function that best represented the data [7]. The number of change points was chosen to be four for the transfer time-series data using leave-one-out cross validation.

The cross-correlation function between weekly transfers and an adjusted weekly number of COVID-19 cases was calculated to understand their associations. For this analysis a variance stabilizing transformation was used on each time-series. Significant cross-correlations (at the

.05 level) were identified as those exceeding approximate 95% confidence intervals (denoted by the dashed lines in Fig 2).

## Results

From April through October 2020, the COVID-19 infection rate was consistently higher in the state's prison population relative to its general population of adults 18 years or older living in the same state. By the end of October 2020, the state had detected over 4,000 COVID-19 cases among its incarcerated population with about 15% of the incarcerated population having tested positive. Beginning in April of 2020, the state detected multiple COVID-19 outbreaks in prisons, which was largely due to symptomatic testing. The state also conducted one-time universal testing during the Summer of 2020, resulting in the detection of many asymptomatic cases.

There were over 20,000 observed transfers between the state's prisons from January through October 2020. The mean age of those transferred was 38.5 years. The majority of transfers occurred among Black (50.7%) followed by White (43.1%) incarcerated persons. Males comprised 96.7% of transfers. This mirrors the demographics within the state's prison system (e.g., 51.5% Black, 39.8% White, 92% Male). The majority of individuals who were transferred (82%), experienced a single transfer, but others experienced multiple transfers, with one individual experiencing nine transfers. Most facilities had similar transfer activities with the exception of processing centers that had the greatest number of transfers. Statewide transfers and admissions into facilities decreased in May 2020 (Fig 1). For the rest of this section, the focus is on weekly transfer data.

### Change in transfer volume due to state-wide policy

Significant changes in the average number of transfers around dates of COVID-19 policy changes (Fig 1) were identified. Weekly transfers decreased the week of April 22, 2020 (95%

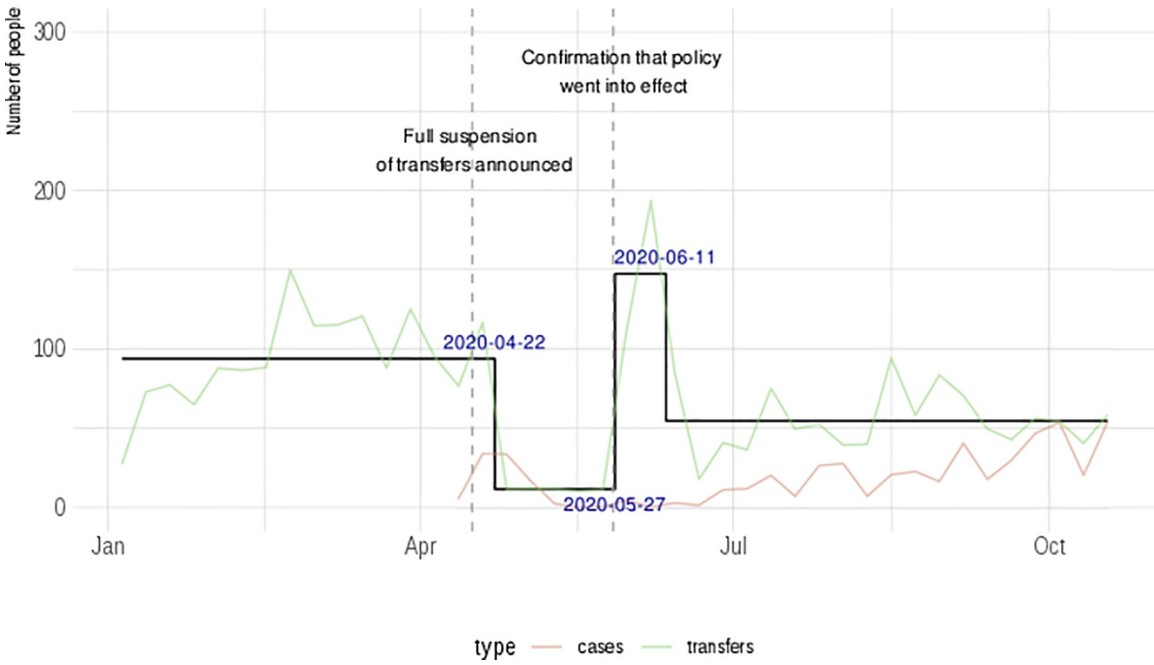

**Fig 1. Number of incarcerated individuals transferred between prisons and number of COVID-19 cases within a single state prison system in 2020.** Green line represents raw weekly average per-day transfers, red line represents raw weekly average per-day COVID-19 cases, black line represents the output of the change point detection model. The two dashed lines represent the timing of a state policy change, and a confirmation of that change at a later date.

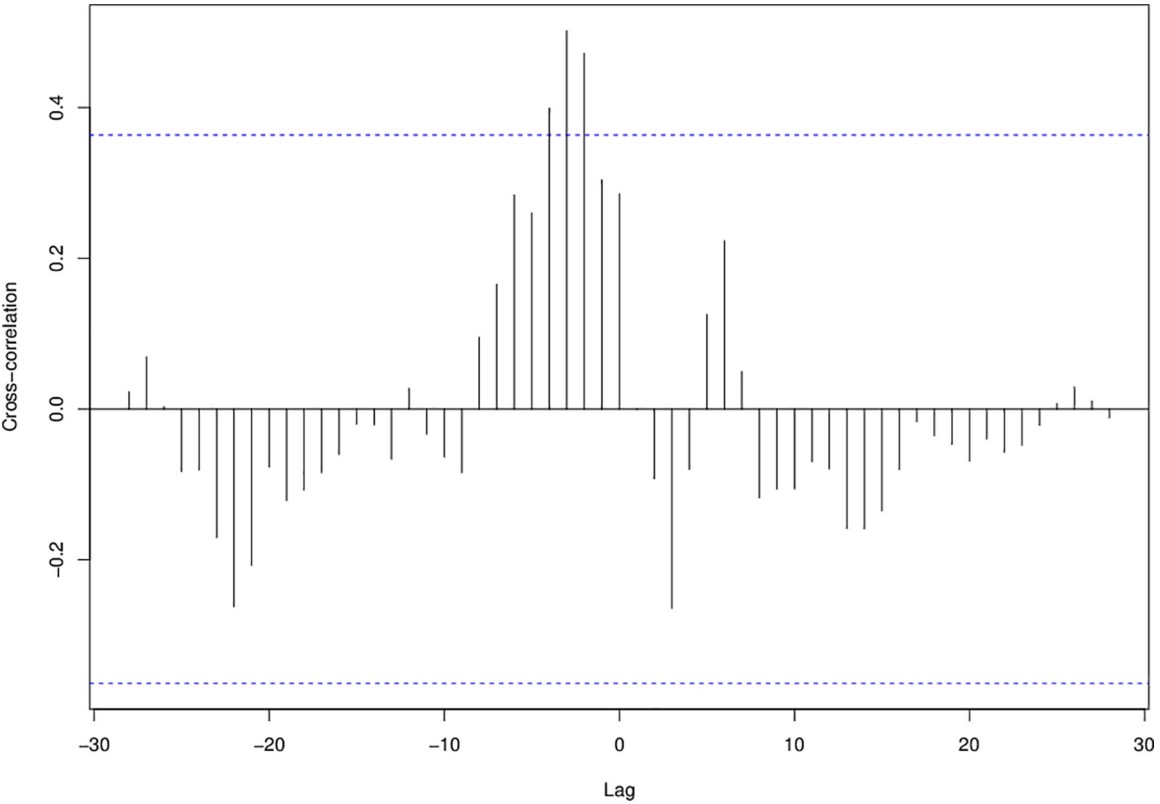

**Fig 2. Cross correlation function between logged values of weekly transfers and COVID-19 positive cases.** A positive correlation at lag -3 to -5 indicates that log transfers are positively associated with log COVID-19 cases 3-to-5 weeks later.

CI: 04/19/20–04/25/20), around the time of an announcement of a statewide order that suspended intersystem transfers. An exception to this reduction happened over the weekend immediately following the announcement when over 500 incarcerated persons were transferred. The average number of transfers dramatically increased the week of May 27, 2020 (95% CI: 05/24/20–05/30/20) immediately prior to the confirmation that suspension of transfer went into effect. The average number of weekly transfers then settled at slightly below original level the week of June 11, 2020 (95% CI: 06/07/20–06/18/20).

## Correlation between the number of transfers and COVID-19 cases

There were significant associations between weekly transfers and positive COVID-19 cases. The number of COVID-19 cases was positively correlated with the number of transfers three to five weeks before (Fig 2, cross-correlations greater than 0.4, p<0.05). In other words, an increase in the number of transfers today was correlated with a significant increase in COVID-19 cases three to five weeks from now. However, an increase in the number of COVID-19 cases today was not associated with a significant change in future transfers.

## Discussion

As the COVID-19 pandemic continues, the case rate growth in carceral institutions continues to outpace that of the general population. While incremental measures have been put in place in prisons in the state to stop the spread of COVID-19 (e.g., testing, quarantine for newly incarcerated individuals), it is unclear to what extent these measures have been carried out and

what effect they have had on COVID-19 spread. In one state's prison system, there was a reduction in the number of transfers early in the pandemic but, subsequently, this decrease was not sustained. The main finding of this study is that transfers were related to future COVID-19 case rates.

In the general population, the primary tool to mitigate COVID-19 spread has been social distancing and, essentially, limiting movement. In carceral settings, incarcerated people have little control over their movements. Most prison facilities are overcrowded, congregate spaces that are vulnerable to COVID-19 outbreaks. Large-scale and continuous transfers of incarcerated people further heighten risk. This risk was widely acknowledged by a majority of prison systems that reported the official prohibition of transfers and new admissions from jails early in the pandemic, including the state system reported on in this analysis. Transfers in prisons can occur for many reasons and are sometimes necessary. For instance, a person may need to be transferred because of medical or safety needs. However, the results of this study demonstrate that, over a 10-month period and at the height of the COVID-19 pandemic, there were over 20,000 transfers. Additionally, there was a large increase in transfers the week of May 27, 2020, which coincided with an increase in COVID-19 transmission. Based on this study's findings, carceral facilities should limit transfers that are not absolutely necessary, especially without other COVID-19 precautions (e.g., testing before and after transfer, mask mandates, ensuring quarantine capacity).

## Limitations

Our analysis was limited to one state due to data availability. As each state's carceral institutions have enacted different COVID-19 policies and mitigation efforts and have experienced different trends in COVID-19 cases, it is important for future work to assess the relationships between transfers and COVID-19 at the national level. Additionally, all data used for this analysis is publicly reported by the state prison system. For COVID-19 data, there are often gaps between testing and reporting of COVID-19 cases. Similarly for transfer data, the authors cannot independently verify the accuracy of these data. When movement designations are not defined, we cannot account for them. There is also not full transparency of data regarding other relevant COVID-19 mitigation measures, such as mandatory masking, social distancing requirements during transfer, and the implementation of such policies. Lastly, while aggregation helps overcome some of the data quality and data sparsity issues it does have potential for ecological bias, as analyses were conducted at the aggregated state level rather than the facility level. Given the relationship between intersystem transfers and COVID-19 cases, there is a need for states to provide transfer data within facilities as well. Multiple states claim to have stopped transfers within facilities (e.g., between units) but more data transparency is needed to understand if these policies are being adhered to across the country.

## Conclusion

Carceral institutions continue to be the epicenter of the COVID-19 pandemic in the United States. Many incremental measures have been put in place to lower COVID-19 spread in prison settings. This was the first empirical analysis to examine the relationship between transfers and COVID-19 incidence in a state prison system. The analysis demonstrated that transfers were positively associated with future COVID-19 cases. To mitigate COVID-19 spread in carceral settings, it is crucial for transfers of individuals between facilities to be limited.

## Acknowledgments

We are grateful to the entire COVID Prison Project team.

## Author Contributions

**Conceptualization:** Lauren Brinkley-Rubinstein, Katherine LeMasters, Phuc Nguyen, Kathryn Nowotny, David Cloud, Alexander Volfovsky.

**Data curation:** Lauren Brinkley-Rubinstein, Alexander Volfovsky.

**Formal analysis:** Phuc Nguyen, Alexander Volfovsky.

**Funding acquisition:** Lauren Brinkley-Rubinstein, Kathryn Nowotny, David Cloud.

**Investigation:** David Cloud.

**Methodology:** Lauren Brinkley-Rubinstein, Phuc Nguyen, Alexander Volfovsky.

**Project administration:** David Cloud.

**Resources:** Kathryn Nowotny.

**Software:** Phuc Nguyen, Alexander Volfovsky.

**Supervision:** Kathryn Nowotny, Alexander Volfovsky.

**Visualization:** Phuc Nguyen.

**Writing – original draft:** Lauren Brinkley-Rubinstein, Katherine LeMasters, Phuc Nguyen, Kathryn Nowotny, David Cloud, Alexander Volfovsky.

**Writing – review & editing:** Lauren Brinkley-Rubinstein, Katherine LeMasters, Phuc Nguyen, Kathryn Nowotny, David Cloud, Alexander Volfovsky.

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
