## [Decision Letter · Decision Letter 0]

2 Aug 2021

The Association between Intersystem Prison Transfers and COVID-19 Incidence in a State Prison System

PONE-D-21-18775

Dear Dr. LeMasters,

We’re pleased to inform you that your manuscript has been judged scientifically suitable for publication and will be formally accepted for publication once it meets all outstanding technical requirements.

Kind regards,

Sungwoo Lim, DrPH

Academic Editor

PLOS ONE

Additional Editor Comments (optional):

1. Line 104, page 5: Please provide a more detailed definition of the general population. For example, are they adults ages 18 years or older living in the same state?

2. Lines 118-119, page 6: Please complete a fragmented sentence.

3. Line 132, page 6: Please insert 95% before CI.

4. Lines 164-165, page 7: Given fluctuation of weekly transfer numbers during 10-month period, readers may not know whether or not ~20,000 transfers are small or large numbers. Instead I suggest highlighting dramatic increase of transfers the week of May 27 which coincides with the peak of the COVID-19 transmission.

Reviewers' comments:

Reviewer's Responses to Questions

**Comments to the Author**

1. Is the manuscript technically sound, and do the data support the conclusions?

Reviewer #1: Yes

2. Has the statistical analysis been performed appropriately and rigorously? 

Reviewer #1: Yes

3. Have the authors made all data underlying the findings in their manuscript fully available?

Reviewer #1: Yes

4. Is the manuscript presented in an intelligible fashion and written in standard English?

Reviewer #1: Yes

5. Review Comments to the Author

Reviewer #1: The manuscript was a well-written, had sound methodology, and adds to the current literature exploring how to mitigate future epidemics/pandemics in correctional settings. There are a few minor recommendations:

• Line 49: Also, correctional officers and staff repeatedly entering and leaving the facility can contribute to the transmission of COVID-19 and spikes in prisons. In fact, several staff have died throughout the country. I don’t think this factor should be ignored and can be added as a quick note.

• Line 163: Can the authors discuss reasons for transfers early? I think it gives the reader a bit more context when you are mentioning the frequency of transfers.

• Figures: Make sure the X and Y axis are clearly labeled on both figures.

6. PLOS authors have the option to publish the peer review history of their article (what does this mean?). If published, this will include your full peer review and any attached files.

Reviewer #1: No

---

## [Editor Report · Acceptance letter]

4 Aug 2021

PONE-D-21-18775 

The Association between Intersystem Prison Transfers and COVID-19 Incidence in a State Prison System 

Dear Dr. LeMasters:

I'm pleased to inform you that your manuscript has been deemed suitable for publication in PLOS ONE. Congratulations! Your manuscript is now with our production department. 

Kind regards, 

on behalf of

Dr. Sungwoo Lim 

Academic Editor

PLOS ONE